# Pilot Study on the Molecular Pathogenesis of Pyeloureteral Junction Obstruction: Underdevelopment or Fibrosis?

**DOI:** 10.3390/medicina59101729

**Published:** 2023-09-27

**Authors:** Ramune Zilinskaite Tamasauske, Vytis Kazlauskas, Povilas Barasa, Natalija Krestnikova, Darius Dasevicius, Vytautas Bilius, Gilvydas Verkauskas

**Affiliations:** 1Clinic of Gastroenterology, Nephrourology and Surgery, Institute of Clinical Medicine, Faculty of Medicine, Vilnius University, 10257 Vilnius, Lithuania; vytis.kazlauskas@santa.lt (V.K.); vytautas.bilius@santa.lt (V.B.); gilvydas.verkauskas@santa.lt (G.V.); 2Institute of Biochemistry, Vilnius University Life Sciences Center, 10257 Vilnius, Lithuania; povilas.barasa@gmc.vu.lt (P.B.); natalija.krestnikova@bchi.vu.lt (N.K.); 3Centre of Pathology, Vilnius University Hospital Santaros Clinics, 10257 Vilnius, Lithuania; darius.dasevicius@vpc.lt

**Keywords:** pyeloureteric junction, pyeloureteric junction obstruction, gene expression, fibrosis-related genes

## Abstract

*Background and Objectives:* Congenital ureteral stenosis is one of the leading causes of impaired urinary drainage and subsequent dilatation of the urinary collecting system, known as hydronephrosis or ureterohydronephrosis. The mechanism that leads to obstruction is not clearly known. Multiple studies in rat models have shown increased angiotensin II and TGFβ levels in obstructed ureteral tissue. The aim of the study is to investigate the expression of fibrosis-related genes in obstructive and normal ureteral tissue. *Material and Methods:* It is a monocentric pilot study in which nineteen patients were selected prospectively. 17 patients underwent Hynes-Anderson pyeloplasty due to the PUJO; two patients underwent ureteroneocystostomy due to ureterovesical junction obstruction (UVJO); and six patients were chosen for the control group: five underwent nephrectomies due to the kidney tumor and one underwent upper pole heminephrectomy due to the duplex kidney with normal pyeloureteric junctions in all. Tissue RNA was chemically extracted after freezing the biopsy samples in liquid nitrogen, with cDNA synthesis performed immediately after nucleic acid isolation. qPCR was performed to evaluate the relative expression of Tgfb1, Mmp1, Timp1, Pai1, Ctgf, and Vegfa. Expression levels of the Gapdh and Gpi genes (geometric average) were used to calculate the relative expression of the investigated genes. Outliers were removed prior to calculating confidence intervals for the experimental groups, and a Wilcoxon rank-sum test was performed to determine the statistical significance of the differences. *Results:* Significant differences between healthy and stenotic tissue samples in Ctgf gene expression levels were observed, with the samples from afflicted tissue showing lower expression. No statistical difference in expression levels of Tgfb1, Timp1, Vegfa, Mmp1, and Pai1 was found. *Conclusions:* These findings suggest that tissue fibrosis, similar to other tissues and organs, is not the leading cause of stenosis, at least at the moment of surgery. Decreased CTGF expression is indicative of the developmental origin of obstruction.

## 1. Introduction

Congenital ureteral stenosis is the leading cause of impaired urinary drainage and subsequent dilatation of the urinary collecting system, and a condition known as hydronephrosis or ureterohydronephrosis, when ureteric dilatation is present. Ureteral obstruction is a very serious clinical condition, leading to the damage of renal parenchyma because of obstruction-induced increased pressure, reduced glomerular filtration, activation of profibrotic factors, cell apoptosis, and subsequent loss of kidney function, which has been investigated in animal models in a series of studies [1,2]. Early recognition of obstruction, followed by timely treatment, usually results in the recovery of a kidney, mainly because of the better reproduction potential of renal stem cells and the recovery of parenchyma at an early age [3].

Several theories of pathogenesis leading to the development of hydronephrosis were learned from animal models. The reasons for hydronephrosis can be generally regarded as obstructive and non-obstructive.

In a systematic analysis that reviewed data from 10 articles, it was revealed that 15 genes had changed expression in ureteral stricture tissue. Genes which were upregulated were (ET1, 87 ACTA2, MCP-1, TGFB1, NFKB1, IL-6, HIF1A, S100A1, SYP), and the expression of six genes was downregulated (ADM, NOS2, EGF, PDGFRA, UCHL1, NGFR). The products of these genes are components of the HIF-1 signalling pathway and participate in the development of vasculature. Some of these factors act as inductors of signalling pathways and particularly affect the Ras signalling system. There may also be a connection between current factors and others that are related to tissue hypoxia, fibrosis, and inflammation [4].

Histological studies of obstructed human ureters identified an excessive amount of collagen in the extracellular matrix, and impaired distribution of its isoforms, an increased proportion of elastin, and apoptosis of myocytes. Altogether it leads to the abnormal development of tissue [5,6]. On the intracellular level, immunofluorescence and PCR analysis revealed that cytoskeleton structural proteins were impaired in PUJO smooth muscle cells, probably explaining the increased apoptosis of smooth muscle cells in PUJO specimens [7]. The aforementioned pathological findings suggest that the pathogenesis of the development of ureteral obstruction involves the processes of fibrosis, apoptosis, and cell differentiation arrest.

The aim of our pilot study was to analyze several fibrosis-related factors that are known to participate in the general mechanism of fibrosis in multiple sites of the human body and to discuss their involvement in previously described chains of pathogenesis. 

We have chosen to investigate fibrosis-associated gene expression in congenitally obstructed ureteral tissue. As obstruction involves abnormal cell differentiation as well as changes in the extracellular matrix, the expression profiles of the *Tgfb1*, *Ctgf*, *Vegfa*, *Timp1*, *Pai1, and Mmp1* genes were chosen to be analyzed according to previous studies where they were analyzed as fibrosis-inducing factors [8,9,10,11,12,13]. We could not identify any studies describing the expression analysis of CTGF, MMP1, and PAI-1 in obstructed ureteral tissue over the last 10 years. Nevertheless, these genes were chosen as closely related to the Tgfb pathway and implicated in the development of tissue fibrosis in other organs, such as the heart [14].

## 2. Materials and Methods

### 2.1. Collection of Samples

It is a monocentric pilot study in which patients were included prospectively from 2019 to 2021. Samples of pyeloureteric junctions and two ureterovesical junctions were collected during surgery and stored at −80 °C until RNA extraction. 

The decision of whether or not to operate on the patient was based on our local protocol: the operation was performed when there was an increase in the anterior-posterior diameter of the renal pelvis by 20% on subsequent ultrasound scans, when the differential renal function (DRF) was <40% of the affected kidney or a decrease in DRF by >5% was observed on subsequent renal scans, or when symptoms could be attributed to hydronephrosis. 

### 2.2. RNA Extraction and qPCR

Tissue RNA was extracted after freezing the samples in liquid nitrogen, mechanically dissociating them, and applying Trizol reagent (Thermo Fisher, Waltham, MA, USA) according to the manufacturer’s protocol. 1 mL of Trizol reagent was added per 100 mg of tissue. Isolated RNA was dissolved in nuclease-free water, and its quantity and quality were evaluated using the NanoDrop device (Thermo Fisher). 

cDNA synthesis was performed using the Maxima H Minus First Strand cDNA Synthesis Kit (Thermo Fisher). Each sample had reverse transcription-negative (RT-) control. qPCR was performed using the Luminaris Color HiGreen qPCR Master Mix (Thermo Fisher) with the Eppendorf realplex4 quantitative PCR device. Expression levels of the *Gapdh* and *Gpi* genes (geometric average) were used to calculate the relative expression of the investigated genes. The common base method, as described in [15], was used to calculate the relative expression of the genes, factoring in the qPCR reaction efficiency. Primer sequences used for qPCR were derived from Primer Bank (https://pga.mgh.harvard.edu/primerbank/ (accessed on 1 October 2022)).

### 2.3. Statistical Analysis

Results were analyzed using RStudio Version 1.2.5033. Outliers were removed prior to analysis. A Wilcoxon rank-sum test was carried out to determine the significance of differences between healthy and obstructed tissue (significance level α = 0.05), and an assessment of confidence intervals was carried out to further understand the differences between the groups.

## 3. Results

There were 17 patients that underwent Hynes-Anderson pyeloplasty due to the PUJO; two patients underwent ureteroneocystostomy due to ureterovesical junction obstruction (UVJO). Six patients were chosen for the control group: five underwent nephrectomies due to the kidney tumor and one underwent upper pole heminephrectomy due to the duplex kidney with normal pyeloureteric junctions in all. The median age was 15.2 months [9.07; 66.2] at the time of the surgery, and the median age of the control group (four boys and two girls) was 60 months [35; 80.3].

The expression of fibrosis-related genes in every specimen is depicted in the heatmap. (Figure 1). Variable expression of all studied genes was noted both in obstructed and normal junctions. The demographic characteristics of patients are shown in the Table 1. 

The Wilcoxon rank-sum test showed significant differences between groups in *Ctgf* gene expression level. We further decided to compare the confidence intervals between groups, and the intervals of the *Ctgf* gene showed little overlap, indicating differences between obstructed and control tissue samples. (Figure 2) No statistically significant differences between the groups were found in the expression of other studied genes.

## 4. Discussion

Out of our selected markers, the best-described group of fibrosis-related factors in the literature belongs to the transforming growth factor (TGF) family, particularly TGFs 1 to 3, sharing similar functions with minor structural differences [16]. Out of the three, the most well-studied and ubiquitous is TGFβ1, a polypeptide that regulates cell differentiation, proliferation, adhesion, and immune behavior. It is found in sites of injury, inflammation, cancer, and elsewhere. The protein both positively and negatively regulates other growth factors. The mature peptide may form heterodimers with other TGF beta family members [17]. It is stored in the extracellular matrix (ECM) as a complex with its prodomain. Activation of TGFβ-1 requires the binding of alpha-V integrin, present on the membrane of residing cells, to an RGD sequence in the TGF prodomain and the exertion of force on this domain, which is held in the extracellular matrix by latent TGFβ binding proteins (LTBP). As the synthesis and release of TGFβ involve many steps of proteolytic processing, binding, and release, the expression and activity of TGFβ are both time- and situation-dependent.

Among the disease-relevant ROS-dependent genomic targets (reactive oxygen species), TGF-beta1 stimulates expression of the potent profibrotic matricellular PAI-1 and CTGF genes as well as those that code ECM structural elements (fibronectin, collagen I). PAI-1 is one of the most highly regulated parts of the TGFβ1/SMAD3-induced group, a prominent ROS-responsive gene involved in the tissue fibrosis process. As the main inhibitor of plasmin generation, PAI-1 limits ECM degradation, alleviating the accumulation of matrix structural elements at the injury site. PAI-1 deficiency is renal-protective, whereas transgenic PAI-1 overexpression promotes an increased fibrotic response with associated recruitment of inflammatory cells, macrophages, and myofibroblasts. As proof-of-concept, unilateral ureter-obstructed PAI-1^−/−^ mice develop a significantly attenuated inflammatory response, suggesting that PAI-1 directly promotes infiltration of macrophages and T-cells [17].

However, it is worth mentioning that out of these studies, we could only identify one when polymerase chain reaction (PCR) analysis was applied in human ureters and showed that the amount of TGFβ1 mRNA in stenotic tissue was higher than in controls [18]. Another study found that immunohistochemical staining of TGFβ3 in obstructed tissue was higher than in the control tissue [19]. Even though the results of our study of TGFβ in patients with PUJO could not replicate these findings, this could be due to the observations that TGFβ1 is often expressed in the early stages of diseases, especially when macrophage activity is high, as well as when inflammation is present. Experimental studies have also advocated the increase in mRNA TGF-β in postoperative UPJO. Under these circumstances, such findings can be explained by an acute onset of obstruction activating muscle fiber and collagen elaboration [20].

We found only one study analyzing the role of PAI-1 in an obstructed kidney model and showing that TIMP-1 and PAI-1 in rat models are involved in the progression of tubulointerstitial scarring of renal tissue [21].

With regard to one of the theories of UB branching, AngII is named as one of the enhancers for this process. PAI-1 has also been described as a downstream target of AngII in endothelial cells [22]. Is there a relation between potentially decreased PAI-1 and AngII, as its potential effector in congenitally obstructed ureter, remains to be answered. PAI-1 has been shown to play a role in ECM maintenance and remodeling, counteracting the effects of matrix metalloproteinases [23].

We were able to identify only one study where VEGF was investigated in UPJO and a control group in human specimens: no color staining of VEGF was found in both obstructed and control groups and subsequently showed no difference in staining potential regardless of obstruction pattern [23]. Our results in VEGF expression have corresponded to the results of an aforementioned study, finding no difference in this gene expression in stenotic versus healthy tissue.

We could not identify any previous publications regarding Ctgf expression profile in congenital ureteral obstruction.

Our results have demonstrated that relative expression of Ctgf in stenotic ureteral tissue was significantly lower than in the control tissue, supporting the potential role of this gene in the processes of stricture development. Ctgf has important roles in many biological processes, including cell adhesion, migration, proliferation, angiogenesis, skeletal development, and tissue wound repair, and is critically involved in fibrotic disease and several forms of cancer [9,24]. It is thought that CTGF can cooperate with TGFβ to induce fibrosis and extracellular matrix production in association with other fibrosis-related conditions [25]. Overexpression of Ctgf in fibroblasts promotes fibrosis at the beginning of the development of stricture in the skin, kidney, and lung. We may only speculate that the reduction in our patients may be indicative of the stage of fibrosis—for example, in a nephropathy model in mice, it was shown that the expression level of Ctgf gradually increases over a two-week period and is static afterwards; our results indicate there may be a significant reduction in the expression of this gene following the primary stage of disease development or that the gene can only be expressed for a certain period of time [26].

Evident similarity in the expression of other studied genes may suggest that the fibrosis process is not the most important in the development of congenital PUJO, and we should rather concentrate on other aspects of development. 

Considering other aspects of cell distribution, the most prominent alteration comprises interstitial cells of Cajal and telocytes, which are significantly reduced in comparison to normal ureteral tissue in both UPJO and UVJO. Differentiation of Cajal cells may be stimulated by overexpression of the CTGF gene [27]. Furthermore, SOX9 was found to regulate CTGF/CCN2 transcription in chondrocytes [28]. Hypomotility of the ureter was induced by *Sox9*, a homeobox gene, the loss of which led to smooth muscle cell differentiation arrest [29]. Another study showed an essential role of *Sox9* in the regulation of differentiation in the ureteric mesenchyme. Primary dilatation of the renal collecting system without ureteral obstruction was induced by *Keap1* knockout and the expression of aquaporins with spontaneous mutations, which caused urinary hyperproduction and secondary dilatation of the renal collecting system [30].

The drawback of our study is the relatively low and heterogeneous age sample size. Although statistical analysis was possible, more specimens would be needed to elucidate the potential differences in the expression of our studied genes. Another drawback is a small and non-homogenous control group when five pyeloureteric junctions were obtained from renal tumor specimens and one pyeloureteric junction was obtained from the duplex kidney system. None of them had macroscopic signs of stricture; however, the results can be influenced by preoperative chemotherapy and maldevelopment of the ureteral wall when compared to normal ureters [31]. Observed tendencies comparing PUJ, UVJ, dilated UVJ, and normal PUJ after chemotherapy may be further investigated.

In summary, our findings suggest that tissue fibrosis, similar to other tissues and organs, is not the leading cause of stenosis, at least at the moment of surgery. Decreased *Ctgf* expression in a strictured ureteral specimen may be a sign of a progressive failure determined by a genetic trigger occurring during the embryonic period. Further studies have to be conducted to verify hypotheses involving wider gene panels, proteomic analysis, and histological verification.

## Figures and Tables

**Figure 1 medicina-59-01729-f001:**
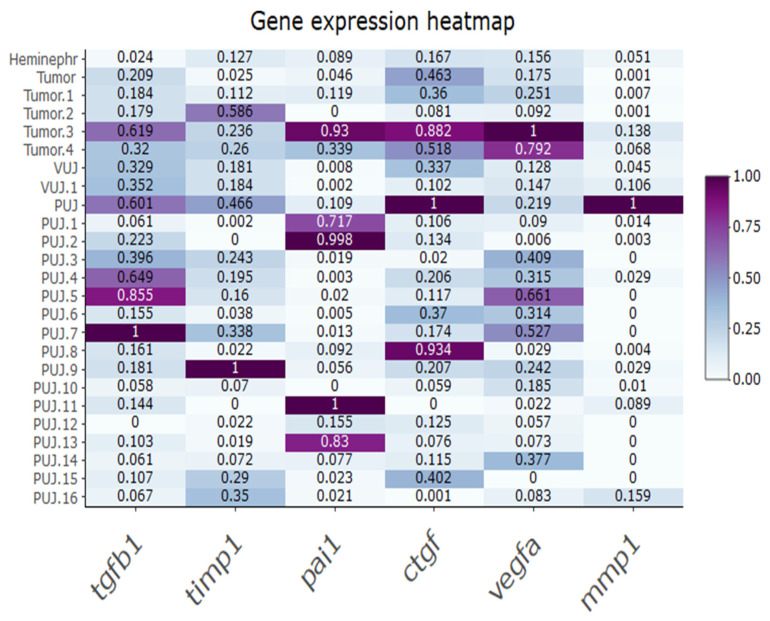
Heatmap of relative expression of fibrosis-related genes in control and strictured ureter tissue samples. Results standardized to highest gene expression levels for each gene. PUJ—pyeloureteric junction obstruction; Tumor—pyeloureteric junction from kidney tumor specimen; Heminephr—pyeloureteric junction from heminephrectomy specimen; VUJ—vesicoureteral junction obstruction.

**Figure 2 medicina-59-01729-f002:**
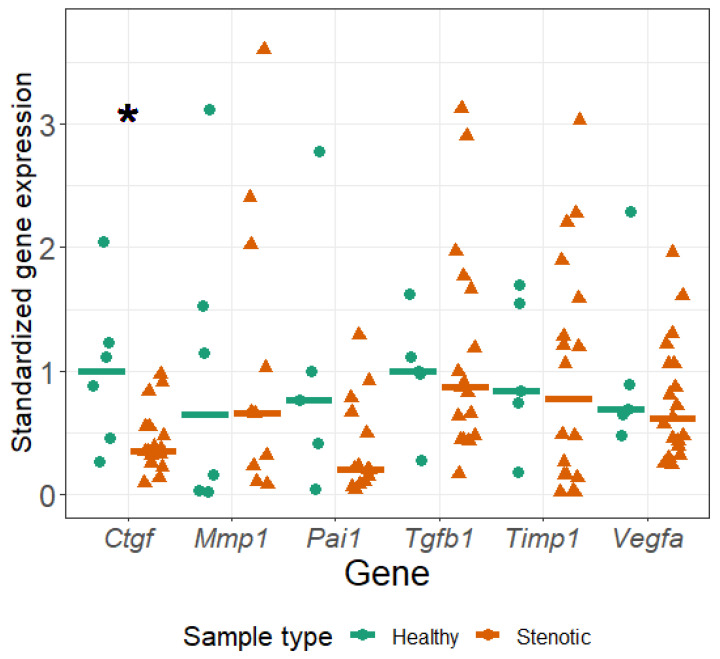
Relative expression of fibrosis-related genes in control and strictured ureter tissue samples. Medians (bars) and datapoints (circles/triangles) shown. Star (*) indicates statistical significance between the groups (*p* < 0.05) as determined by Wilcoxon rank-sum test. Sample sizes are 6 for control and 19 for stenotic (strictured) group, outliers were removed according to interquartiliary range method.

**Table 1 medicina-59-01729-t001:** Demographic characteristics of patients.

Sample	Age at Enrollment (Months)	Sex (1-Boy, 0-Girl)	Weight (kg)	Height (cm)
Heminephr.	10.l37	1	n/a	n/a
Tumor	7.47	1	8.7	73
Tumor.1	98.13	1	28.5	132
Tumor.2	24.87	1	n/a	n/a
Tumor.3	68.73	1	n/a/	n/a
Tumor.4	2.90	1	5.4	64
VUJ	3.33	1	n/a	n/a
VUJ.1	29.60	n/a	n/a	n/a
PUJ	21.43	0	12.8	79
PUJ.1	54.83	1	22	114
PUJ.2	1.40	0	4.8	55
PUJ.3	51.33	n/a	n/a	n/a
PUJ.4	3.97	1	7.24	68
PUJ.5	111.33	0	35	141
PUJ.6	9.53	0	7.5	71
PUJ.7	95.43	1	33.5	142
PUJ.8	21.50	1	14.5	89
PUJ.9	3.77	1	8.9	71
PUJ.10	199.73	n/a	n/a	n/a
PUJ.11	84.10	n/a	n/a	n/a
PUJ.12	11.13	1	13.5	78.5
PUJ.13	98.60	0	24.6	127
PUJ.14	3.63	1	7.63	64
PUJ.15	78.80	1	27	126
PUJ.16	2.6	0	5.3	56

n/a: not applicable.

## Data Availability

Not applicable.

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
