# Peer review of "Pilot Study on the Molecular Pathogenesis of Pyeloureteral Junction Obstruction: Underdevelopment or Fibrosis?"

_medicina, 2023, doi:10.3390/medicina59101729_

Round 1

Reviewer 1 Report

The present manuscript investigates several fibrotic agents associated with genes expression in congenitally obstructed ureteral tissue. The subject is very important with the references in general up to date and the language seems correct. Previous studies have advocated the increase expression of TGFβ-1 in postoperative UPJO that might explain an acute obstruction due to muscle fiber and collagen elaboration. TIMP-1, PAI-1, and VEGF-A play a crucial role in ECM remodeling, counteracting the effects of MMPs. The interesting finding of this study suggests that decreased CTGF expression in strictured ureteral specimen may be a sign of a progressive failure determined by a genetic trigger occurring at the embryonic period.

However, the results may be influenced by preoperative chemotherapy and maldevelopment of the ureteral wall when compared to normal ureters. The relatively low and heterogeneous by age sample size with the number specimens may lead to lack of elucidation of the potential differences in the expressed studied genes. Another limitation is the small non-homogenous control group with 5 pyelo-ureteric junctions obtained from renal tumor.

I'd like to see a revised form of the present study improving these limitations in order to verify the hypothesis of tissue fibrosis as the leading cause of stenosis. 

Author Response

Response 1: We have changed the concept of the study to the “pilot study” and corrected the title and discussion accordingly. Sample size was limited to the period of time accorded to the study of this rare anomaly prospectively and to the number of eligible patients particularly the controls. The baseline project of this study is finished and the presented analysis of results are in favor of different project methodology in the future. We think that results would be of interest for researchers planning similar studies.

Thank you.

Reviewer 2 Report

The debated topic is interesting.

The Authors want to investigate expression of fibrosis related genes in congenital obstructive and normal ureteral tissue.

They decided to investigate the expression of 6 genes, poorly analyzed in the past.  The patients were 21 : 17 with obstructed pyeloureteric junction, 2 with uretherovescical junction obstruction and 6 with normal pyeloureteric junction (5 with kidney tumor and 1 with upper pole heminephrectomy due to duplex kidney).

I think this paper is not suitable for publication due to the following reasons:

- The control group is not with normal ureteral tissue as there are patients who underwent nephrectomy for cancer with or without prior therapy that was not specified and that could had an impact on the expression of the selected genes;

- As the authors indicated, the groups are not homogenous by age and sample size;

- The results of the study showed a significative difference between groups in CTGF gene expression level. The authors blamed it on the hypothetical regressive stage of fibrosis and concluded that “decreased ctgf expression in strictured ureteral specimen may be a sign of a progressive failure determined by a genetic trigger occurring at the embryonic period”. Yet it is known that CTGF gene if overexpressed in different tumors, one of them is Wilms tumor that is very likely to be this case.

non

Author Response

Thank you for your review.

The control group is not with normal ureteral tissue as there are patients who underwent nephrectomy for cancer with or without prior therapy that was not specified and that could had an impact on the expression of the selected genes; As the authors indicated, the groups are not homogenous by age and sample size; The results of the study showed a significative difference between groups in CTGF gene expression level. The authors blamed it on the hypothetical regressive stage of fibrosis and concluded that “decreased ctgf expression in strictured ureteral specimen may be a sign of a progressive failure determined by a genetic trigger occurring at the embryonic period”. Yet it is known that CTGF gene if overexpressed in different tumors, one of them is Wilms tumor that is very likely to be this case.

Respond1: Thank you for your insights. Together with a response to your third question we have changed discussion and conclusions, and referenced it. As to control group - it is impossible to operate on not diseased ureters and to get absolutely normal control group in the research topic. In our opinion, having disclosed the origin of control group, our findings are important for future research. Similarly, to the second remark: most of pyeloplasties are performed at the earliest in life to preserve kidney function. „Controls“ are operated later for poorly functioning kidneys or tumors.

Reviewer 3 Report

The aim of the study is to investigate expression of fibrosis related genes in obstructive and normal ureteral tissue. 

Major comments

What is the novelty of your work?

How did you calculate the sample size?

The title is not in accordance with the findings.

It is suggested to add a table on demographic characteristics.

Minor comments

There are many grammatical errors such as angiotensin 11,... . Please revise it in the whole part of the manuscript.

All abbreviations should be described for the first use. 

Sentences should not be initiated with numerics.

Author Response

What is the novelty of your work?

Response 1: Prospective molecular study of a rare congenital anomaly with controls. Control human, especially pediatric, tissue is hardly available because of ethical issues. Some of our investigated genes (Pai1, Timp1, Mmp1) have been studied in the context of other types of fibrosis (lungs, heart, among others), but not in the case of ureteral stenosis.

How did you calculate the sample size?

Response 2: This prospective multicentric study was licensed for a limited period of time, and we obtained a maximum available during that period. As a pilot it seems enough to direct future studies.

The title is not in accordance with the findings.

Response 2: We corrected the title

It is suggested to add a table on demographic characteristics.

 Response 3: added in the article.

There are many grammatical errors such as angiotensin 11,... . Please revise it in the whole part of the manuscript.

Response 4: fixed in the text

All abbreviations should be described for the first use. 

Response 5: Corrected where found. Main abbreviations are described separately. The majority of discussed genes are not described but referenced as not being the subject of this study.

Sentences should not be initiated with numerics.

Response 6: corrected in text.

Thank you.

Reviewer 4 Report

The authors have developed an interesting work on Pyeloureteral junction obstruction in patients, which is of great value for studying pathology. The manuscript is coherent and well-structured, but it needs to be improved in different aspects. To this end, I suggest the following corrections:

Abstract: Angiotensin 11? Probably Angiotensin II.

Due to the small number of patients, I recommend including the "Pilot study" concept.

"In systematic analysis which reviewed data from 10 articles was revealed that genes had changed expression in ureteral stricture tissue." What systematic analysis was performed, and how was it achieved? Is it an already published article?

Methods: Indicate the number of patients included in the study and the inclusion and exclusion criteria. Subsequently, in the results, make a table indicating patient characteristics (age, sex, clinical variables...).

Statistical analysis: What method has been used to eliminate outliers?

The figures are out of order (figure 2 has been included first).

Figure 2: Please make a graph with dots of each patient's values to better visualize the result obtained.

The manuscript has essential limitations: a small number of patients and a small experimental part. qPCR is an interesting and correct technique, but in basic research, different experimental approaches are usually performed to explore the working hypothesis. I suggest including histological experiments that identify fibrosis per se in the tissue and immunohistochemistry.

The grammatical quality of the manuscript is acceptable

Author Response

Point 1: Due to the small number of patients, I recommend including the "Pilot study" concept.

Response 1: corrected accordingly.

Point 2: "In systematic analysis which reviewed data from 10 articles was revealed that genes had changed expression in ureteral stricture tissue." What systematic analysis was performed, and how was it achieved? Is it an already published article?

Response 2: Yes, it is published: reference 4.

Point 3: Methods: Indicate the number of patients included in the study and the inclusion and exclusion criteria. Subsequently, in the results, make a table indicating patient characteristics (age, sex, clinical variables...).

Response 3: Indicated accordingly including table 1: the number of patients included in the study was 26. One patient was excluded due to the sample defect (necrolysis of the sample). Patients were included in the study having and operated due to the ureter stenosis (PUJ or VUJ).

Point 4: Statistical analysis: What method has been used to eliminate outliers?

Response 4: Added: outliers were removed by applying the interquartiliary range (IQR) exclusion method. Briefly, all gene expression values that did not fit within [Q1 - 1,5 x IQR; Q3 + 1,5 x IQR], Q1 and Q3 being first and third quartile, respectively, were excluded for each gene in the control and fibrotic groups.

Point 5: The figures are out of order (figure 2 has been included first).

Response 5: corrected in the article.

Point 6: Figure 2: Please make a graph with dots of each patient's values to better visualize the result obtained.

Response 6: corrected in the article.

Point 7: The manuscript has essential limitations: a small number of patients and a small experimental part. qPCR is an interesting and correct technique, but in basic research, different experimental approaches are usually performed to explore the working hypothesis. I suggest including histological experiments that identify fibrosis per se in the tissue and immunohistochemistry.

Number of patients is limited because of rare anomaly and ethical issues with controls. Thank you for good ideas in the future. We did not include histology because of routine findings of fibrosis on previously routine pathological studies. Immunohistochemistry is missing and we acknowledge that.

Round 2

Reviewer 1 Report

Published in present revised form.

Reviewer 2 Report

none

Reviewer 3 Report

My comments have been responded by the authors.

 Minor editing of English language required

Reviewer 4 Report

Thank you for following my suggestions. I think the article has improved, although you should consider improving the aesthetics of Figure 2.

Kind regards